# Home Environment in Early-Life and Lifestyle Factors Associated with Asthma and Allergic Diseases among Inner-City Children from the REPRO_PL Birth Cohort

**DOI:** 10.3390/ijerph191911884

**Published:** 2022-09-20

**Authors:** Katarzyna Kapszewicz, Daniela Podlecka, Kinga Polańska, Iwona Stelmach, Pawel Majak, Barbara Majkowska-Wojciechowska, Bogusław Tymoniuk, Joanna Jerzyńska, Agnieszka Brzozowska

**Affiliations:** 1Department of Pediatrics and Allergy, Copernicus Memorial Hospital, Medical University of Lodz, Pabianicka Street 62, 93-513 Lodz, Poland; 2Department of Environmental and Occupational Health Hazards, Nofer Institute of Occupational Medicine, 91-348 Lodz, Poland; 3Poddębice Health Center, 99-200 Poddębice, Poland; 4Department of Pediatric Pulmonology, Copernicus Memorial Hospital, Medical University of Lodz, 93-513 Lodz, Poland; 5Department of Immunology and Allergy, Medical University of Lodz, 93-513 Lodz, Poland

**Keywords:** HDM allergy, asthma, children, atopic dermatitis, allergic rhinitis

## Abstract

Objective. We hypothesized that, in our REPRO_PL cohort, exposure to indoor allergens and lifestyle factors in early life are associated with risk of asthma, atopic dermatitis, and allergic rhinitis at ten years of age. Methods. We only examined children who had lived in the same house from birth. Children’s exposure to tobacco smoke was assessed based on cotinine levels in urine. House dust samples were collected. Results. Higher Fel d1 concentration in house dust was associated with significantly higher risk of developing asthma at age 10 years (95% CI,10.87 to 20.93; *p* < 0.001). Frequent house cleaning was associated with development of atopic dermatitis (odds ratio 0.61; 95% CI 0.37 to 0.99; *p* = 0.045). Clustering of exposure to HDM revealed two types of environment. Cluster 1, defined as lower HDM (dust), in contrast to Cluster 2, defined as higher HDM, was characterized by old-type windows, lower fungus and dampness levels, as well as more frequent house cleaning. Conclusion. Exposure to cat allergens and new-type buildings that limit air flow while increasing the condensation of steam on the windows and thereby stimulating the growth of fungi are risk factors for the development of asthma.

## 1. Introduction

Environmental exposures in early life play an important role in the development of childhood asthma and allergic diseases, but the etiologic pathway is complex and dynamic [1]. Childhood allergy is associated with sensitization to inhalant allergens, while the impact of early-life exposure to these allergens, and their sources, on the risk of asthma, atopic dermatitis (AD), and allergic rhinitis (AR) are unclear and differ between allergens [2,3,4,5]. Exposure to animal allergens constitutes a relevant risk factor for the development of allergic sensitization and respiratory allergic diseases in susceptible individuals. Homes with pet dogs or cats have a distinct microbiome and mice orally supplemented with house dust from homes with dogs are protected against ovalbumin or cockroach allergen-mediated airway pathology [6,7]. Other potentially modifiable factors in urban populations that may be linked to childhood allergies and respiratory health include tobacco smoke and housing conditions, such as the presence of indoor molds, home dampness, and the type of windows and heating system [8,9,10,11]. The results of European birth cohorts showed that maternal smoking during pregnancy was related to asthma in preschool children whose mothers did not smoke postnatally [12]. Moisture damage and mold at an early age in the child’s main living areas have been shown to be associated with the risk of developing persistent asthma and respiratory symptoms during the first 6 years [9].

The Polish Mother and Child Cohort (REPRO_PL) was designed to investigate risk factors for allergic diseases such as asthma, AD, and AR in a birth cohort of high-risk inner-city children, a population with a high burden of asthma morbidity and a distinct environment. As our study is a prospective one, we observed this cohort from birth. We have collected a long questionnaire concerning environmental exposures and children’s condition from birth up to the moment of analysis. We only examined children who had lived in the same house from birth. Current conditions are maintained from the first moments of life of the children. In a previous report on our REPRO_PL cohort, the associations between different environmental factors affecting pregnant women and young children (aged 1–2 years old) and children’s health status were identified [13,14,15,16].

The aim of the current study is to evaluate the associations between a broad range of early childhood environmental exposures—including indoor allergen concentration, tobacco exposure, and housing conditions—and asthma and allergic diseases in school children from the birth cohort.

## 2. Materials and Methods

### 2.1. Study Design and Population

The present study is based on data from the Polish Mother and Child Cohort (REPRO_PL) follow-up until 10 years of age. REPRO_PL is an ongoing multicenter prospective cohort study established in 2007 [17,18]. The mothers’ recruitment, follow-up procedures, and complete description of the methodological assumptions have been published elsewhere [17,18]. Women were interviewed during pregnancy and gave biological samples for assessment of the level of exposure to environmental factors. After the child’s birth, mothers and children participating in the REPRO_PL cohort were followed to monitor the child’s health status and environmental exposure to factors such as tobacco.

The present study was conducted between 2018 and 2019, ten years after the child’s birth, and was restricted to children (and their mothers) who had lived in the same house from birth. An invitation letter for a doctor’s visit was sent to mothers, including an offer to participate in follow-up examinations covering environmental exposure (including questionnaire data) and child health examination (details below). The house dust samples from the children’s home were collected following the doctor’s visit.

All the mothers of the patients gave their written consent before the study. The study was approved by the Ethical Committee of the Nofer Institute of Occupational Medicine, Łódź, Poland (Decision No. 7/2007, 3/2008 and 22/2014) for mothers and children up to 8 years of age and by the Medical Ethics Committee of the Medical University of Lodz (Decision No. RNN/388/17/KE) for older children.

### 2.2. Child Health Assessment

Children’s health status was assessed at the time of the doctor’s visit. For the appropriate assessment of children’s health status, a questionnaire was administered to the mothers and supplemented with information from the medical chart of each child. This part of the questionnaire was developed by an allergist, based on recommendations from the International Study of Asthma and Allergies in Childhood (ISAAC), and had been applied previously [18,19,20]. In addition, the occurrence of allergies among family members was noted. The clinical examination was performed by a pediatrician/allergist in the presence of the mother or a relative. All patients underwent skin prick tests and allergy was confirmed or excluded. The diagnosis of asthma, allergic rhinitis, or atopic dermatitis was confirmed by a physician at the time of the interview.

### 2.3. Allergic Sensitization

Skin prick testing for standard allergen extracts (e.g., house dust mite, cat, dog, cockroach, mouse, and Alternaria) was performed using Allergopharma, Reinbek, Germany. Reactions >3 mm in diameter above the negative control recorded at 15 min were considered positive.

### 2.4. Allergic Morbidity

Atopic dermatitis diagnosis was established by a physician’s clinical examination, and disease severity was assessed using the SCORAD questionnaire [21]. We considered the severity of AD as mild (SCORAD index under 15), moderate (SCORAD between 15 and 50), and severe (SCORAD above 50).

Asthma was evidenced by a doctor’s diagnosis and based on respiratory symptoms such as wheezing, tightness of the chest, and/or asthma attack; or medication intake such as bronchodilators, inhaled/oral steroids; or specific oral treatment improving the child’s breathing according to GINA guidelines. The reversibility test was performed when necessary [22].

Allergic rhinitis was assessed on the basis of a doctor’s diagnosis and with reference to symptoms such as sneezing or runny/blocked nose or rhino-conjunctivitis when the child did not have a cold or the flu according to ARIA [23].

### 2.5. Assessment of Children’s Exposure to Environmental Factors

The relevant exposure data were assessed based on questionnaires during interviews. The questionnaire covered demographic and socioeconomic information, medical and reproductive history, and the home environment. All potential confounding factors regarding parents and children, such as parental status, were evaluated. Among the environmental factors distinguished in this study were conditions and type of domicile—tower block, tenement building, detached house; type of windows—new-type (plastic, PCV), old-type (wooden); presence of a pet animal in the house (none, dog, cat, dog and cat, rodents); presence of humid and/or fungal areas in the house. House dust samples to measure allergen levels were collected once after the interview. Details regarding the covariates are presented in Table 1. The child’s exposure to environmental tobacco smoke was assessed based on cotinine levels in urine collected from children aged 10 years at the time of the interview (doctor’s visit).

### 2.6. Cotinine Levels in Urine

These were analyzed using high-performance liquid chromatography coupled with tandem mass spectrometry/positive electrospray ionization (LC-MS/MS-ESI+) and the isotope dilution method. A description of the methodological assumptions has been published elsewhere (as described by Stragierowicz et al. [24] and Lupsa et al. [25]).

### 2.7. Collection and Analysis of Dust Samples

House dust samples were collected once. They were obtained by trained research personnel using a KIRBY (model Sentria II) vacuum (The Kirby Company, Cleveland Ohio, OH, USA) with a special dust collector (Micromagic) fitted into the inlet hose of the vacuum. House dust samples were collected from the floors of the bedroom. Floors were vacuumed for at least 5 min, with particular attention to the corners of the room. Dust samples were weighed and extracted by adding natural saline buffered by PBS-tween (0.05%). Two milliliters of buffer were used for every 100 mg of dust. If the sample weight was lower, the amount of buffer was added proportionately. Then, sites were mixed and shaken for 2 and 1/2 h (on a roller mixer) at room temperature; next, they were whirled for 20 min/4C/250 rpm. Supernatants were frozen and stored in the Biobank at the Immunology and Allergy Clinic of Medical University of Lodz in Eppendorf test tubes at −20 °C until the procedure for allergen concentration measurement started.

All dust samples were analyzed by the 7-plex Multiplex Array method, using a Luminex 100/200 TM device based on the Maria test manufactured by INDOOR Biotechnologies Inc. (Wiltshire, United Kingdom) that measured common indoor allergens: house dust mite allergens Der p 1 (Dermatophagoides pteronyssinus), Der f 1 (Dermatophagoides farinae), animal allergens Can f 1 (dog), Fel d 1 (cat, Felis domesticus), Mus m 1 (mouse, Mus musculus), German cockroach, Bla g 2 (Blattella germanica), and mold allergen Alt a 1 (Alternaria alternata). The lower limits of detection were <0.06 ng/mL for Der p 1, Der f 1, Can f 1, Bla g 2; <0.02 ng/mL for Fel d 1 and Alt a 1; and <0.006 ng/mL for Mus m 1. Each extract was added to a color-coded magnetic ball mixture, initially covered with antibodies specific for each analyte. The antibodies were connected with their specific analytes. Then, biotinylated detecting antibodies, peculiar for each analyte, were administered to create an antibody–antigen structure. In the next step, streptavidin conjugated with phycoerythrin (PE) was added to connect with biotinylated detecting antibodies. After all these procedures, magnetic balls were read on a dual-laser based on the detector flow in a Luminex 100/200 TM analyzer. First, the laser classifies balls and describes the detectable analyte; second, the laser defines the rate of signal from PE, which is in direct proportion to the concentration of combined analytes. The concentration of examined allergens was given in ng/mL.

All measurements were performed by the Asthma and Allergy Patients Support Organization at the Medical University of Lodz, Poland (Immunology and Allergy Clinic).

### 2.8. Statistical Analysis

The relationships of demographic, perinatal, family, and environmental factors to asthma, atopic dermatitis, and allergic rhinitis at the age of 10 years were assessed using logistic regression models. The relationship between environmental exposures and the atopic diseases was assessed using a logistic regression model adjusted for sociodemographic factors. The significance of differences in mean allergen concentrations in dust was appraised using multifactor generalized linear models with robust standard errors (due to the non-normal distributions of the assessed data). Allergen and other environmental measurement data were log-transformed prior to analysis.

Clustering analyses and between-cluster comparisons of clinical data have been planned to determine the possible link between indoor exposure to allergens and house condition and allergic outcomes in school children. Allergen concentrations were included in the cluster analysis. Box–Cox transformation was performed followed by clustering using the expectation–maximization (EM) algorithm. The number of clusters was selected according to the results of v-fold cross-validation. Between-groups comparisons were performed using the Mann–Whitney test for continuous data and the two-tailed Fisher’s or Pearson χ2 tests for categorical data. Holm–Bonferroni correction for multiple comparisons was used. Differences were considered statistically significant at a *p*-value of 0.05 or less. Statistica 13.1 (TIBCO Software Inc., Palo Alto, CA, USA) was used to perform all analyses.

## 3. Results

### 3.1. Description of the Study Cohort

The current analysis is restricted to 103 children, and their mothers who had lived in the same house from birth and were being followed-up, who had participated in a health examination at ten years of age. The lifetime prevalence of the selected outcomes including home environmental factors is given in Table 1. About 33% of children had a positive parental history of atopy. Pets were kept in 54% of homes. The level of parental education was high (64.71% of mothers and 36.27% of fathers had completed university-level education). In 31.37% homes, at least one parent was a smoker. A frequency of house cleaning of more than twice a week was reported in 43.14% of homes. Of those enrolled children, 16 (15.53%) had asthma, 29 (28.16%) had atopic dermatitis, and 28 (27.18%) were sensitized to one or more aeroallergens and had allergic rhinitis at age 10 years.

#### 3.1.1. Indoor Allergen Exposure and Health Outcomes

The most frequent allergens detected in the homes of the studied children were cat and dog allergens, detected in 100% of them, followed by mouse allergen and house dust mite Der p 1 (Table 2). The highest concentration of allergens was for dog and cat (36.41 ng/mL and 11.69 ng/mL, respectively). Cockroach allergen was detected only in 5 houses, with a very low mean concentration of the allergen (0.11 ng/mL); therefore, this was not analyzed further.

Data regarding the concentrations of allergens in house dust and selected home environmental factors are presented in Table 3.

For the aeroallergen Fel d 1, a higher allergen concentration in house dust was associated with a significantly higher risk of developing asthma at age 10 years, with a geometric mean of 15.09 (95% CI, 10.87 to 20.93; *p* < 0.001) (Table 4). The non-significant association of higher allergen exposure with higher risk of developing asthma was observed among all participants. As for AD and AR, no significant associations were observed.

#### 3.1.2. Association between Environmental Factors and Health Outcomes

In both univariate and multivariate analyses, children whose fathers were older were more likely to develop asthma and allergy (odds ratio 0.83; 95% CI 0.72 to 0.96; *p* = 0.011 and odds ratio 0.87; 95% CI 0.82 to 0.98; *p* = 0.022, respectively). Frequent house cleaning was associated with the development of atopic dermatitis (odds ratio 0.61; 95% CI 0.37 to 0.99; *p* = 0.045). Other demographic factors and perinatal history were not associated with asthma, allergic rhinitis, or atopic dermatitis occurrence (Table 1). Neither urine cotinine concentration nor parental smoking were predictors of asthma or any of the other studied diseases. No other factors underlying the investigated diseases in the studied children were associated with the development of these diseases.

#### 3.1.3. Cluster Analysis

Clustering of exposure to HDM revealed two clinically distinct types of environment (Table 5, Figure 1). Cluster 1, defined as lower HDM, included 67% (*n* = 69) of our cohort. The environment of Cluster 1 was characterized by old-type windows, lower fungus, and lower dampness levels as well as more frequent cleaning (*p* < 0.0001). Cluster 2, defined as higher HDM, included 33% (*n* = 34) of our cohort. The environment of Cluster 2 was characterized by new-type windows, higher fungus, and higher dampness concentrations as well as less frequent cleaning (*p* < 0.0001).

## 4. Discussion

The REPRO_PL cohort study evaluated the home environments of children in urban neighborhoods beginning in the prenatal period to identify potentially modifiable risk factors for childhood asthma and other allergic diseases. Higher exposure to cat allergens during infancy was associated with a higher risk of developing asthma at an age of 10 years. Indoor house dust concentrations of allergens other than cat were not associated with the development of asthma in this cohort. In addition, cotinine concentration was not a risk factor for asthma at age 10. Clustering of exposure to HDM in children revealed two clinically distinct types of environment. Cluster 1, defined as lower HDM (concentration in dust)—in contrast to Cluster 2, defined as higher HDM—was characterized by old-type windows, lower fungus, and lower dampness levels as well as more frequent house cleaning.

Exposure to environments rich in allergens and microbes has been associated with low rates of asthma and other allergic diseases [8,25,26,27]. It has been shown that the environment of Amish farms in the USA is associated with a lower prevalence of asthma and allergen sensitization than Hutterite farms, and house dust from the former setting has a higher endotoxin concentration and distinct microbiota compared with the latter [28]. Compared with other environments, inner cities are extensively paved and may have a paucity of outdoor exposures to animals, plants, and sources of microbes such as soil and green space. In this setting, indoor pets may provide biological signals that promote normal lung and immunological development. In our research, we have proved that higher exposure to cat allergens during infancy is associated with a higher risk of developing asthma at an age of 10 years, but we have not found similar results for other allergens. Research by O’Connor et al. [27] based on children up to 7 years of age showed that higher concentrations of cockroach, mouse, and cat allergens in house dust in the first three years of life were associated with a lower risk of asthma (for cockroach allergen, the odds ratio for an increase in the interquartile range 0.55; 95% CI, 0.36 to 0.86; *p* < 0.01). At the same time, the abundance of a number of bacterial taxa in house dust was associated with increased or decreased asthma risk [29]. On the other hand, it is known that diesel pollution and exhaust fumes intensify the action of allergens, which may affect the development of allergic inflammation [30,31]. The majority of our investigated patients lived in big cities. Furthermore, in cluster 2 patients, mold and dampness were much higher, which is also reported to be linked with asthma [32]. This is especially interesting when considering these results in the context of Lu et al. [33], who stated that exposure to tobacco smoke, dampness and mold, condensation on windows in wintertime, and keeping cats and dogs were associated with furry pet-related symptoms such as rhinitis or wheezing. At the same time, cooking on an electric stove and early-life exposure to animals (cats, dogs, farm environment during pregnancy) were protective for diagnosed rhinitis not related to furry pets [33].

Relationships between allergen exposure and wheezing or asthma could depend on the nature of the allergen exposure and the other environmental factors present in a specific environment; for example, exposure to dust mite in URECA was not associated with asthma risk, while in another US study early-life dust mite exposure was associated with increased asthma risk [27,34]. Further, in contrast to our findings for cat allergen in inner cities, a pooled analysis of data from 11 European birth cohorts found no association between the presence of cats in homes during infancy and asthma at school age, although those studies did not include measured allergen concentrations [35]. At the same time, our findings are consistent with those in US inner cities [27,34].

The most surprising finding in our research was the fact that living in old buildings with old, leaky windows that allow more airflow was associated with a lower risk of developing allergies and asthma. Perhaps the fact that the airflow is greater reduces the likelihood of dampness and mold growth, factors that may contribute to the development of allergic diseases. The second unexpected finding of our research was that neither urine cotinine concentration nor parental smoking were predictors of asthma or any of the other studied diseases. In other studies [27,29], parental smoking and cotinine levels were strong predictors of asthma development. Nevertheless, just 31% of investigated families reported exposure to tobacco smoke. Furthermore, in some cases, parents smoked only outside and not inside the home; thus, we could have false results.

In our study, we also found that frequent house cleaning was associated with development of atopic dermatitis. This result is consistent with the hygienic theory of the development of allergic diseases; moreover, it seems to be a consequence of the destruction of the skin’s lipid layer by detergents and cleaning agents. These findings are also consistent with the results of Huy Ta et al., who stated that exposure to household environmental microbiota during critical periods in early life may be a risk factor for eczema development in early childhood [29]. Huy Ta et al. demonstrated an inverse relationship between the beneficial bacteria of animal origin and bacteria associated with humans, which protects against atopy and eczema. The absence of these interactions in the eczema groups may indicate a lack of inhibitory effects by the beneficial environmental bacteria Planomicrobium on human-associated bacteria. The presence of only human-related microflora in places of residence can be harmful and predispose children to the development of allergies, in the absence of regulation by the natural environmental microbiota [29].

The strengths of this study include a definition of asthma based on the observations of a physician’s diagnosis, symptoms, medication use, and lung testing. However, a limitation is that the relationships with environmental factors may be specific to this population and may depend on the personal or environmental context, including specific allergies, lifestyle, or neighborhood factors. We did not measure the gastrointestinal or airway microbiome in early childhood; so, we cannot evaluate the independent effects of the house dust, airway, and gastrointestinal microbiomes. Despite the prospective and longitudinal design of the study, with environmental assessments beginning at 10 years of age, it is possible that a family’s health and health-related behaviors lead to long-term alterations of home allergens, microbiome, and maternal stress, rather than these factors influencing health outcomes.

## 5. Conclusions

In summary, our study showed that exposure to cat allergens increases the risk of the development of asthma in children. In addition, new buildings that limit air flow while increasing the condensation of steam on the windows and thereby stimulating the growth of fungi is a risk factor for the higher concentration of house dust mite allergens in the place of living, and thereby the development of asthma and atopic dermatitis. More information is needed to identify specific exposures that could potentially be modified.

## Figures and Tables

**Figure 1 ijerph-19-11884-f001:**
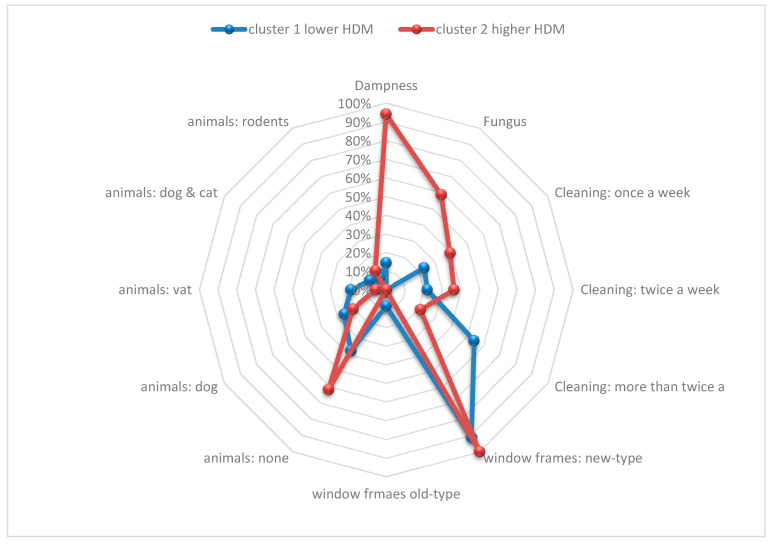
Environmental exposure characteristics according to cluster analysis.

**Table 1 ijerph-19-11884-t001:** Baseline characteristics of the study cohort (environmental and family factors) (*n* = 103).

Factor	No. of Patients/Mean	Percentage/SD
Domicile:		
Tower block	18	17.48
Block of flats	37	35.92
Tenement building	11	10.68
Stand-alone house	37	35.92
Window frames:		
New-type	97	94.17
Old-type	6	5.83
Domestic animals:		
None	47	45.63
Dog	25	24.27
Cat	15	14.56
Dog and cat	7	6.80
Rodents	9	8.74
Cockroaches	2	1.94
Dampness	42	40.78
Fungus	20	19.42
Male patients	45	43.69
Mother’s age (years)	29.27	4.49
Father’s age (years)	30.91	5.22
Mother’s education:		
Primary	4	3.92
Secondary	32	31.37
Higher	66	64.71
Father’s education:		
Primary	4	3.92
Secondary	61	59.81
Higher	37	36.27
Childbirth:		
Natural spontaneous	55	53.40
Natural induced	6	5.82
Forceps or vacuum	0	0.00
Caesarean section	42	40.78
Birth weight (g)	3343.11	380.23
Breast-feeding	84	81.55
Familial atopy	34	33.33
Smoking	32	31.37
Building age (years)	25.74	13.13
Heating system:		
Central, oil, or gas	78	76.47
Coal, coke, tiled stove	24	23.53
Carpet surfaces	47	46.08
Cleaning:		
Less than once a week	2	1.96
Once a week	29	28.43
Twice a week	27	26.47
More than twice a week	44	43.14
Cotinine concentration in urine (pg/mL)	1.91	2.59

**Table 2 ijerph-19-11884-t002:** Descriptive statistics for the investigated allergen prevalence (%) and concentrations (ng/mL) in the studied children.

Allergen	No. of Patients	Percentage	Geometric Mean	95% CI
*Alt a 1*	55	53.40	0.0153	0.0102–0.0204
*Bla g 2*	5	4.85	0.11	0.04–0.28
*Can f 1*	103	100.00	36.41	25.78–51.41
*Der f 1*	86	83.50	0.70	0.50–0.97
*Der p 1*	93	90.29	1.60	1.16–2.21
*Fel d 1*	103	100.00	11.69	8.63–15.84
*Mus m 1*	102	99.03	0.56	0.42–0.74

**Table 3 ijerph-19-11884-t003:** Descriptive statistics for the concentrations of allergens in dust (ng/mL) in the studied children by selected environmental factors.

Factor	*Alt a 1*		*Can f 1*		*Der f 1*		*Der p 1*		*Fel d 1*		*Mus m 1*	
	Geometric Mean	95% CI	Geometric Mean	95% CI	Geometric Mean	95% CI	Geometric Mean	95% CI	Geometric Mean	95% CI	Geometric Mean	95% CI
Tower block	0.0152	0.0074–0.0312	50.45	27.05–94.09	0.38	0.22–0.65	1.03	0.51–2.07	19.87	8.35–47.31	0.67	0.30–1.51
Block of flats	0.0196	0.0139–0.0275	44.12	22.20–87.68	0.63	0.37–1.09	1.45	0.81–2.58	18.13	9.68–33.95	0.38	0.24–0.60
Tenement building	0.0192	0.0031–0.1197	17.69	5.34–58.57	0.24	0.11–0.50	2.44	1.19–5.02	3.93	2.41–6.42	0.51	0.28–0.92
Stand-alone house	0.0173	0.0102–0.0294	31.78	18.38–24.94	1.43	0.79–2.57	1.91	1.04–3.52	8.05	5.89–10.99	0.76	0.46–1.25
*p*-value	=0.268		=0.259		<0.001		=0.019		<0.001		=0.175	
New-type windows	0.0172	0.0133–0.0222	37.50	26.25–53.57	0.71	0.51–0.99	1.59	1.14–2.23	12.32	8.97–16.91	0.53	0.43–0.77
Old-type windows	0.0301	0.0019–0.4819	22.61	3.46–147.81	0.49	0.05–4.68	1.80	0.55–5.88	5.01	1.82–13.78	0.37	0.14–0.94
*p*-value	=0.031		=0.124		=0.992		=0.190		<0.001		=0.024	
Any pet	0.0178	0.0124–0.0257	64.11	39.85–103.15	0.60	0.40–0.90	1.48	0.97–2.24	15.91	10.91–23.20	0.60	0.41–0.89
None pet	0.0180	0.0120–0.0272	18.56	11.90–28.93	0.83	0.48–1.45	1.75	1.06–2.90	8.09	4.98–13.15	0.51	0.33–0.77
*p*-value	=0.946		<0.001		=0.056		=0.283		=0.030		=0.219	
Dog	0.0152	0.0098–0.0237	153.72	76.27–309.81	0.69	0.38–1.24	1.53	0.73–3.20	8.40	6.39–11.04	0.52	0.30–0.90
Cat	0.0258	0.0102–0.0650	17.07	8.65–33.68	0.44	0.19–1.05	1.98	0.84–4.64	44.09	18.31–106.13	1.20	0.48–3.01
Dog and cat	0.0314	0.0054–0.1815	92.69	11.96–718.20	0.95	0.19–4.75	0.90	0.42–1.95	37.54	5.68–248.05	0.33	0.09–1.19
Rodents	0.0066	0.0057–0.0076	38.48	20.84–71.05	0.40	0.11–1.49	1.23	0.34–4.39	8.80	5.02–15.43	0.50	0.16–1.53
*p*-value	=0.342		=0.001		=0.101		=0.093		<0.001		=0.584	
Dampness (+)	0.0187	0.0125–0.0280	36.11	20.53–63.52	0.79	0.46–1.36	2.31	1.33–4.02	10.09	6.41–15.89	0.63	0.39–1.01
Dampness (–)	0.0175	0.0122–0.0251	36.62	23.40–57.29	0.63	0.42–0.97	1.23	0.84–1.80	12.93	8.54–19.59	0.51	0.36–0.73
*p*-value	=0.639		=0.620		=0.381		=0.030		=0.207		=0.256	
Fungus (+)	0.0102	0.0057–0.0181	40.47	19.10–85.74	1.41	0.53–3.75	4.76	1.90–11.91	9.38	4.14–21.23	0.87	0.41–1.83
Fungus (–)	0.0207	0.0154–0.0277	35.49	23.90–52.71	0.59	0.42–0.81	1.24	0.90–1.69	12.33	8.86–17.14	0.50	0.37–0.68
*p*-value	=0.020		=0.941		=0.004		=0.001		=0.370		=0.097	

**Table 4 ijerph-19-11884-t004:** Descriptive statistics for the concentrations of allergens in dust (ng/mL) in the studied children by selected health factors (*n* = 103).

Factor	*Alt a 1*		*Can f 1*		*Der f 1*		*Der p 1*		*Fel d 1*		*Mus m 1*	
	Geometric Mean	95% CI	Geometric Mean	95% CI	Geometric Mean	95% CI	Geometric Mean	95% CI	Geometric Mean	95% CI	Geometric Mean	95% CI
Asthma (+)	0.0188	0.0096–0.0367	43.98	16.71–115.74	1.11	0.40–3.09	1.82	0.77–4.29	15.09	10.87–20.93	0.77	0.41–1.44
Asthma (–)	0.0178	0.0132–0.0240	35.17	24.14–51.23	0.64	0.45–0.91	1.57	1.10–2.23	11.15	7.81–15.92	0.53	0.38–0.72
*p-value*	=0.608		=0.423		=0.416		=0.266		<0.001		=0.255	
Atopic dermatitis (+)	0.0173	0.0094–0.0319	33.80	15.06–75.85	0.94	0.43–2.04	1.55	0.82–2.93	12.34	7.50–20.28	0.62	0.38–1.01
Atopic dermatitis (–)	0.0181	0.0134–0.0246	37.49	25.77–54.55	0.62	0.44–0.88	1.63	1.12–2.37	11.44	7.81–16.77	0.54	0.38–0.76
*p-value*	=0.590		=0.153		=0.226		=0.399		=0.879		=0.500	
Allergic rhinitis (+)	0.0199	0.0136–0.0290	37.54	20.64–68.28	0.69	0.44–1.10	1.74	1.08–2.82	11.83	8.02–17.45	0.53	0.35–0.80
Allergic rhinitis (–)	0.0162	0.0110–0.0240	35.41	23.91–52.45	0.70	0.43–1.13	1.49	0.96–2.31	11.56	7.21–18.53	0.58	0.40–0.86
*p-value*	=0.591		=0.069		=0.645		=0.437		=0.960		=0.670	

**Table 5 ijerph-19-11884-t005:** Environmental exposure characteristics by clustering results.

	Cluster 1 Lower HDM		Cluster 2 Higher HDM		Total		P(chi2)
Domicile:							0.6825
Tower block	11	16%	7	21%	18	17%	
Block of flats	24	35%	13	38%	37	36%	
Tenement building	9	13%	2	6%	11	11%	
Stand-alone house	25	36%	12	35%	37	36%	
Window frames:							0.0314
new-type	63	91%	34	100%	97	94%	
old-type	6	9%	0	0%	6	6%	
Domestic animals:							0.0338
none	26	38%	21	62%	47	46%	
dog	18	26%	7	21%	25	24%	
vat	13	19%	2	6%	15	15%	
dog and cat	7	10%	0	0%	7	7%	
rodents	5	7%	4	12%	9	9%	
Dampness	10	14%	32	94%	42	41%	<0.0001
Fungus	0	0%	20	59%	20	19%	<0.0001
ETS	25	37%	6	18%	31	30%	0.1420
Heating system:							0.2926
central, oil, or gas	49	72%	29	85%	78	76%	
coal, coke, tiled stove	19	28%	5	15%	24	24%	
Carpet surfaces	28	41%	19	56%	47	46%	0.1306
Cleaning:							0.0022
less than once a week	0	0%	1	3%	1	1%	
once a week	16	24%	13	39%	29	29%	
twice a week	15	22%	12	36%	27	27%	
more than twice a week	37	54%	7	21%	44	44%	
Atopic dermatitis	17	24.6%	12	35.3%	29	28.2%	*p* = 0.35147
Asthma	8	11.6%	8	23.5%	16	15.5%	*p* = 0.10147
Atopy	32	46.4%	17	50.0%	49	47.6%	*p* = 0.83444
grass	28	87.5%	15	88.2%	43	87.8%	*p* = 1.0000
dog	4	12.5%	6	35.3%	10	20.4%	*p* = 0.07531
cat	13	40.6%	2	11.8%	15	30.6%	*p* = 0.05220
HDM	16	50.0%	10	58.8%	26	53.1%	*p* = 0.38749
molds	4	12.5%	4	23.5%	8	16.3%	*p* = 0.42308

## Data Availability

Not applicable.

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
