# Peer review of "Home Environment in Early-Life and Lifestyle Factors Associated with Asthma and Allergic Diseases among Inner-City Children from the REPRO_PL Birth Cohort"

_ijerph, 2022, doi:10.3390/ijerph191911884_

Round 1

Reviewer 1 Report

Dear Authors,

this is a very interesting manuscript. I suggest you check the linguistic manuscript again.

Author Response

Dear Reviewer, 

thank you for your comments on the paper. The language was corrected according to your suggestion by a native. 

The abstract has been pasted back into the form; earlier, there was a
technical problem that was solved. Sorry for the inconvenience.

Reviewer 2 Report

The authors in the present manuscript relate the presence of allergic pathology such as asthma, dermatitis or rhinitis with different variables the type of housing, the presence of pets ......... They present a number of results that are obvious (all of table 3), such as that homes with pets have a higher incidence of dog and cat allergens.

As very important issues when assessing the manuscript:

1.- The manuscript lacks abstract.

2.- The main objective of the study as stated by the authors in the last paragraph of the introduction “was to evaluate the association between a broad range of  early childhood environmental exposures”. The design of the study is not adequate since the results are obtained at the time of the clinical study of the children. The work reflects the current effect of the parameters studied, not the previous one. It is possible that the authors assume that the current conditions are maintained from the first moments of life of the children, but this should be clearly reflected in the manuscript.

3.- The authors should justify why they use the geometric mean instead of the arithmetic mean for variables such as allergen concentrations that are not percentages.

4.- The most evident clinical finding is the association between high Fel d1 concentrations and asthma. However, the authors did not analyze by skin tests to what degree asthma is induced by cat airborne allergens and how many of the asthmatic children lived with cats.

Author Response

Dear Reviewer,

thank you for your comments on the paper. The language was corrected according to your suggestion by a native. We also corrected some of our references and discussed the results and study design for clarity. All corrections are marked in red. 

  1. 1 1.- The manuscript lacks abstract.

Ans: The abstract has been pasted back into the form; earlier, there was a
technical problem that was solved. Sorry for the inconvenience

Q 2: 2.- The main objective of the study as stated by the authors in the last paragraph of the introduction “was to evaluate the association between a broad range of  early childhood environmental exposures”. The design of the study is not adequate since the results are obtained at the time of the clinical study of the children. The work reflects the current effect of the parameters studied, not the previous one. It is possible that the authors assume that the current conditions are maintained from the first moments of life of the children, but this should be clearly reflected in the manuscript.

  Ans: The present study is based on data from the Polish Mother and Child Cohort (REPRO_PL) follow-up until 10 years of age. REPRO_PL is an ongoing multicenter prospective cohort study established in 2007The Polish Mother and Child Cohort (REPRO_PL) was designed to investigate risk factors for allergic diseases such as asthma and allergic rhinitis in a birth cohort of high-risk inner-city children, a population with a high burden of asthma morbidity and a distinct environment. As our study is a prospective one, we observed this cohort from birth. We have collected a long questionnaire concerning environmental exposures and children’s condition from birth up to the moment of analysis. We only examined children who had lived in the same house from birth. In conclusion, current conditions are maintained from the first moments of life of the children. We added this explanation to the text.

Q 3.- The authors should justify why they use the geometric mean instead of the arithmetic mean for variables such as allergen concentrations that are not percentages.

Ans: When dealing with allergen concentrations that did not meet the assumptions the assumptions about the normality of the distribution and had a remarkably large dispersion along with multiple outliers, it was advisable to compute the geometric mean instead of the weighted arithmetic mean. The Authors’ decision was also supported by previous studies of the scientific literature in the field of allergology, where many authors, in similar protocols, did use the geometric mean.

4.- The most evident clinical finding is the association between high Fel d1 concentrations and asthma. However, the authors did not analyze by skin tests to what degree asthma is induced by cat airborne allergens and how many of the asthmatic children lived with cats.

Ans:

These research questions were beyond the scope of the present paper, at the time. However, we do intend to investigate these areas in the near future which will result in studies encompassing an extended range of analyzes and conclusions.

Reviewer 3 Report

Since the prevalence of allergic diseases increases, identifying their risk factors is paramount. Unfortunately, the published data on that issue are inconsistent. Thus, each article on that subject is valuable, particularly those prospectively investigating well-described birth cohorts, including analysis of house dust mite components, as in the presented manuscript.

The authors analyzed a cohort of 103 children living in large cities in the same housing environment for ten years since birth.

The main outcome is that living in old buildings with old-type windows and extended airflow conditions was a protective factor for developing allergies and asthma. Furthermore, higher cat allergen concentration in house dust was associated with a significantly higher risk of asthma at age 10. It was not true for other allergens, including house dust mites. Moreover, frequent house cleaning was associated with the development of atopic dermatitis. On the other hand, parents’ smoking was not a risk factor for allergy and asthma. However, smoking status was assessed based on cotinine levels in children's urine at the age of 10 years, not analyzing the history of smoking (Is that true?).

The presented results are interesting, the study is well-desing, the manuscript is well-written, and the statistics seem to be correctly done (although I am not a statistician).

 I have only one comment: there is no Abstract in the main submission – it must be corrected.

Minor remark:

Children with older fathers were more likely to develop asthma and allergy. How would the authors explain that association, e.g., by the impact of other cofactors? Please add a short comment on that issue to the Discussion section. 

Author Response

Dear Reviewer, 

thank you for your comments on the paper. The language was corrected according to your suggestion by a native. 

The abstract has been pasted back into the form; earlier, there was a
technical problem that was solved. Sorry for the inconvenience

Q: Children with older fathers were more likely to develop asthma and allergy. How would the authors explain that association, e.g., by the impact of other cofactors? Please add a short comment on that issue to the Discussion section. 

R: In our study indeed children with older fathers were more likely to develop asthma and allergy. The available literature has repeatedly assessed the risk of inheriting asthma in children from their parents. According to the results from metanalysis: 

Lim RH, Kobzik L, Dahl M. Risk for asthma in offspring of asthmatic mothers versus fathers: a meta-analysis. PLoS One. 2010 Apr 12;5(4):e10134. 

 children of asthmatic fathers are more likely to develop asthma than those of non-asthmatic fathers (summary OR 2.44, 95%CI: 2.14–2.79). The test for heterogeneity showed a non-significant p-value of 0.06.  We didn’t find any literature concerning fathers’ age and the risk of asthma in offspring. The study population was too small to be able to analyze the relationship between the age of the fathers and the risk of developing asthma in their offspring, so we decided not to comment on this result for fear of speculation with the data.

Round 2

Reviewer 2 Report

The authors have adequately answered the four questions put to them. They have inserted the abstract in the manuscript. They have clarified that data collection is carried out from birth. They explain the use of geometric mean due to the great dispersion of the data (it should be cited in the text). Finally, they will analyze the effect of Fed d1 in further investigations.